# Natural Product Heme Oxygenase Inducers as Treatment for Nonalcoholic Fatty Liver Disease

**DOI:** 10.3390/ijms21249493

**Published:** 2020-12-14

**Authors:** David E. Stec, Terry D. Hinds

**Affiliations:** 1Department of Physiology & Biophysics, Center for Cardiovascular and Metabolic Diseases Research, University of Mississippi Medical Center, 2500 North State Street, Jackson, MS 39216, USA; 2Department of Pharmacology and Nutritional Sciences, University of Kentucky, 760 Press Avenue, Healthy Kentucky Research Building, Lexington, KY 40508, USA

**Keywords:** NAFLD, NASH, bilirubin, biliverdin reductase, HO-1, HO-2, ROS, inflammation, hepatic function

## Abstract

Heme oxygenase (HO) is a critical component of the defense mechanism to a wide variety of cellular stressors. HO induction affords cellular protection through the breakdown of toxic heme into metabolites, helping preserve cellular integrity. Nonalcoholic fatty liver disease (NAFLD) is a pathological condition by which the liver accumulates fat. The incidence of NAFLD has reached all-time high levels driven primarily by the obesity epidemic. NALFD can progress to nonalcoholic steatohepatitis (NASH), advancing further to liver cirrhosis or cancer. NAFLD is also a contributing factor to cardiovascular and metabolic diseases. There are currently no drugs to specifically treat NAFLD, with most treatments focused on lifestyle modifications. One emerging area for NAFLD treatment is the use of dietary supplements such as curcumin, pomegranate seed oil, milk thistle oil, cold-pressed *Nigella Satvia* oil, and resveratrol, among others. Recent studies have demonstrated that several of these natural dietary supplements attenuate hepatic lipid accumulation and fibrosis in NAFLD animal models. The beneficial actions of several of these compounds are associated with the induction of heme oxygenase-1 (HO-1). Thus, targeting HO-1 through dietary-supplements may be a useful therapeutic for NAFLD either alone or with lifestyle modifications.

## 1. Introduction

Nonalcoholic fatty liver disease (NAFLD) has reached epidemic levels in the United States in the patient population, where it has had a profound effect on the healthcare system [1,2,3]. NAFLD is characterized by hepatic lipid accumulation and insulin resistance that, when coupled with increased inflammation and oxidative stress, can lead to nonalcoholic steatohepatitis (NASH), liver fibrosis, or liver cancer [1]. NAFLD is also associated with a greater risk of cardiovascular disease; however, the mechanism by which NAFLD enhances this risk is not known [1,2,4]. It is becoming recognized that NAFLD is a complex phenotype, as evidenced by the wide spectrum of disease severity as well as the wide variability in the patient presentation of the disease. It has been suggested that NAFLD progresses through multiple “hits” which couple insulin resistance and alterations in adipose tissue lipolysis with increasing efflux of free fatty acids from adipose to the liver resulting in hepatic steatosis. Given this propensity of NAFLD to occur with obesity and other metabolic disorders, renaming the disease metabolic dysfunction-associated fatty liver disease (MAFLD) has recently been proposed [5,6]. Hepatic steatosis makes the liver vulnerable to additional “hits” such as inflammation and increased oxidative stress resulting in hepatocyte injury and disease progression. 

Heme oxygenase (HO) is the rate-limiting enzyme responsible for the breakdown of heme into biliverdin, carbon monoxide, and free heme [7]. Biliverdin reductase reduces biliverdin to bilirubin, and ferritin sequesters the free heme released by HO. Heme is a known generator of cellular oxidative stress but also plays an important role in numerous cellular functions. Heme is a ligand of nuclear receptors of the REV-ERB (α and B) family, where it regulates such processes such as circadian rhythms, metabolism, and immune function [8,9,10]. The HO isozymes are present in two isoforms, HO-1, which is inducible by a wide variety of stimuli, including hypoxia, oxidative stress, exposure to toxins, and natural products [11,12], and HO-2, which is the constitutively expressed isoform. Chemical induction or genetic overexpression of HO-1 protects against numerous cardiovascular diseases such as hypertension, stroke, atherosclerosis, and kidney disease [13,14,15,16,17,18,19]. Human patients with a deficiency in HO-1 present with severe hemolysis, inflammation, and ectopic iron deposition, similar to the phenotypes described in HO-1 null mice [20,21,22,23]. Recent studies have revealed several natural herbal supplements improve NAFLD and metabolic dysfunction, which might be mediated by their induction of HO-1. We discuss these further and the impact that herbal supplements that induce HO-1 may have in relieving the metabolic distress to alleviate the cost burden on the healthcare system. 

## 2. Heme Oxygenase and Hepatic Steatosis

Alterations in hepatic HO levels profoundly affect hepatic steatosis in several rodent models of fatty liver [24,25,26,27]. The most commonly studied method for inducing HO-1 has been with cobalt protoporphyrin IX (CoPP), which is thought to inactivate the BTB domain and CNC homolog 1 (BACH1) transcription factor that inhibits *Hmox1* gene activity [28]. CoPP increases mRNA and protein levels that also equates to higher HO activity measured by higher bilirubin levels [24,29]. Other metal-containing metalloprotoporphyrins can also increase HO-1 expression. However, some increase expression but inhibit activity, such as tin protoporphyrin IX (SnPP) and stannic mesoporphyrin (SnMP). These are the most common pharmacological agonist of HO activity (CoPP) and antagonists (SnMP and SnPP). Studies in obese leptin-deficient (ob/ob) mice treated with CoPP or SnMP showed that both increased HO-1 expression [24]. However, only CoPP decreased plasma heme and increased serum bilirubin levels, which are indicators of raised HO activity. The CoPP-induction of HO activity reduced body weight and blood glucose, as well as improved NAFLD and hepatic glycogen content [24]. These indicate that increasing HO activity may have metabolic protection that could be useful for treating these disorders. 

Both chemical and genetic induction of HO-1 reduces hepatic steatosis in Zucker fatty rats and mice fed high fat or methionine-choline deficient (MCD) diet [24,25,26]. Hepatic induction of HO-1 reduces not only hepatic steatosis but also decreases inflammatory factors such as suppressor of cytokine signaling-1, interleukin-6, and tumor necrosis factor-alpha (Figure 1). HO-1 induction also decreases hepatic macrophage-chemoattractant-protein-1 (MCP-1), with similar reductions in the pro-inflammatory M1-phenotype marker, Anti-Macrophages/Monocytes Antibody, clone ED-1, and resulting in decreased infiltration of hepatic macrophages [25,26]. Likewise, HO-1 induction concomitantly potentiates the protein expression of anti-inflammatory macrophage-M2-markers, including CD-206, ED2, and interleukin-10 (IL-10) [26]. One of the biggest challenges in treating obesity is reducing the chronic-low-grade inflammation associated [30]. 

HO-1 induction not only modifies the inflammatory response in the liver but also alters the level of several genes involved in hepatic fatty acid metabolism, such as fatty acid synthase (FAS) and peroxisome proliferator-activated receptor-α (PPARα) (Figure 1) [24]. Hepatic HO-1 induction also increases fibroblast growth factor-21 (FGF21) levels [24]. FGF21 is an important metabolic hormone that improves insulin sensitivity and decreases hepatic steatosis [31]. HO-1 increases PPARα and FGF21 through enhanced production of bilirubin. Recent studies have identified bilirubin as a novel selective PPARα modulator (SPPARM) [32,33]. Bilirubin is able to alter specific transcriptional coregulators when it is bound to PPARα as a ligand, which increases the levels of PPARα-target genes [32,34,35]. Thus, the effect of HO-1 induction to alter PPARα and its target genes like FGF21 is likely due to enhanced bilirubin production. The bilirubin induction of FGF21 was confirmed to be by PPARα in global PPARα knockout mice treated with bilirubin, which showed no change in expression, but wild-type control mice treated with bilirubin did have higher FGF21 [32]. Later studies confirmed that direct PPARα-bilirubin binding by fluorescent competitive ligand-binding assays [34]. The study also showed by chromatin immunoprecipitation (ChIP) assays that PPARα was enriched at gene promoters and that coregulator recruitment to purified mouse and human PPARα protein was enhanced by treatment with bilirubin [34]. Another study showed an extensive analysis using RNA-sequencing in human HepG2 hepatocytes with lentiviral shRNA knockdown of PPARα that ~95% of bilirubin-controlled gene activity is PPARα-dependent [36]. Treatment of mice with obesity-induced NAFLD with bilirubin nanoparticles reduced hepatic steatosis and improved liver function, lowering AST liver dysfunction marker [35], suggesting that HO-1 protection against NALFD may be mediated by bilirubin. HO production of bilirubin may serve a hormonal function by the actions of bilirubin directly binding and activating the PPARα nuclear receptor transcription factor [37].

Inhibition of HO activity results in increased hepatic lipid accumulation and fibrosis in both animal and cell culture models of NAFLD [25,27]. Inhibition of HO-1 also lowers serum FGF21 levels and reduces the levels of PPARα-target genes in the liver [24]. One mechanism by which blockade of hepatic HO-1 results in enhanced lipid accumulation and fibrosis is through increases in reactive oxygen species (ROS) levels. Enhanced ROS production and reduced antioxidant capacity are emerging areas that have been demonstrated to contribute to enhanced lipid accumulation and fibrosis [38,39,40,41]. Raffaele et al. have shown that epoxyeicosatrienoic intervention increases HO-1- PGC1α-mitochondrial signaling to improve NAFLD in obese mice [42]. The HO-1 activation leads to increased bilirubin-PPARα stimulation, which has also been shown to improve mitochondrial function [34]. Mainly since the loss of hepatic PPARα induces NALFD, which worsens on a high-fat diet [43], these suggest that diet modification with HO-1 induction may have synergism that further improves NAFLD. Also, the inclusion of exercise may benefit this regimen as rats with high aerobic running capacity have been shown to have significantly higher bilirubin levels [44]. 

The regulation of HO-1 gene transcription occurs through a combination of transcription factors like nuclear factor (erythroid-derived 2)-like-2 (Nrf2), which increases HO-1 levels as well as BACH1, which suppresses HO-1 transcription. BACH1 competes with Nrf2 for binding to the Maf recognition elements (MAREs) in the HO-1 promoter [45,46]. Studies in cultured liver cells demonstrated that inhibition of BACH1 levels with specifically targeted siRNAs increases HO-1 levels [47]. Mice that were genetically deficient in BACH1 exhibit reduced levels of hepatic steatosis and preserved levels of PPARα in a methionine-choline deficient (MCD) diet model of NAFLD [48]. DHA (docosahexaenoic acid) is an omega-3 fatty acid contained in high levels in several species of cold-water fish. DHA has a positive effect on a wide variety of diseases such as adult-onset diabetes mellitus, arthritis, atherosclerosis, depression, hypertension, myocardial infarction, and thrombosis. Studies by Wang et al. demonstrated that DHA attenuates BACH1 binding to the Antioxidant Response Elements (AREs) in the HO-1 gene promoter by reducing nuclear BACH1 protein expression by promoting its degradation [49]. In addition, DHA enhances HO-1 gene transcription through Nrf2 binding to the AREs independent of nuclear Nrf2 expression levels [49]. Thus, targeting HO-1 transcriptional repressors such as BACH1 could be a novel therapeutic approach for NAFLD treatment. 

Natural product induction of Nrf2 is another mechanism that may be protective against NAFLD. Mice lacking Nrf2 are highly susceptible to the development of NASH when they are challenged with a methionine- and choline-deficient (MCD) diet. Nrf2 deficient mice showed a substantial increase in hepatic steatosis, hepatic oxidative stress, an increase in hepatic NF-kappaB p65 protein, and an increase in inflammatory markers tumor necrosis factor-α (TNFα), cyclooxygenase 2, and inducible nitric oxide synthase as compared to livers from wild-type mice [50]. These results are in contrast with results from hepatocyte-specific Nrf2 knockout mice place on a high-fat diet. In these mice, the reduction in the levels of peroxisome proliferator-activated receptor γ (PPARγ) and its target lipogenic genes resulted in the attenuation of high-fat diet-induced hepatic steatosis and inflammation [51]. In separate studies, hepatocyte-specific Nrf2 KO mice fed a HFD exhibited improved insulin sensitivity, lower insulin levels and no significant liver triglyceride accumulation [52]. Interestingly, adipocyte-specific knockout of Nrf2 resulted in several metabolic abnormalities including glucose intolerance, high fasting glucose levels, and alteration in plasma fatty acids and cholesterol [52]. The results from the tissue-specific Nrf2 KO studies demonstrate the complex interactions of loss of Nrf2 from different tissues and suggest that specific targeting of Nrf2 in selective tissues may be required to improve metabolic abnormalities associated with diseases like obesity and NAFLD [53]. Studies with natural Nrf2 inducers such as sulforaphane and its precursor glucoraphanin as well as glycycoumarin (GCM) which is a representative coumarin compound in licorice have demonstrated the hepatoprotective effects of these compounds in both models of NAFLD as well as NASH [54,55]. These products induce Nrf2 to improve antioxidant defense, stimulate 5’ AMP-activated protein kinase (AMPK)-mediated energy homeostasis, induce autophagy degradation process, and improve mitochondrial dysfunction. Additional studies in Nrf2 deficient animals are needed in order to determine if these products act through pathways independent of Nrf2 to protect against NAFLD. 

## 3. Natural Product HO-1 Inducers and NAFLD

### 3.1. Curcumin

Curcumin, the bioactive component found in the spice turmeric, has a long tradition in Eastern medicine where it has been used as an anti-inflammatory and antioxidant agent [56]. Curcumin is a potent inducer of HO-1 in the liver [57,58]. Both curcumin or a more bioavailable derivative, pegylated curcumin, have been demonstrated to attenuate the development of NAFLD and the progression to NASH in animal models [59,60]. Curcumin supplementation protects against NAFLD via several distinct mechanisms. Curcumin has potent anti-inflammatory properties through attenuation of markers of inflammation, such as elevated C-reactive protein (CRP) and interleukin-6 (IL-6) levels [61]. Curcumin can also act through the Nfr2/FXR/LXRalpha pathway to regulate the expression of CYP3A and CYP7A as well as alter the expression of CD36, sterol regulatory element-binding protein-1c (SREBP-1c), and FAS to regulate cholesterol and fatty acid metabolism in NAFLD [62]. Hepatic proteomic analysis in methionine and choline-deficient (MCD) mice supplemented with curcumin revealed inhibition of O-linked beta-N-acetylglucosamine (O-GlcNAc) modifications resulting in the attenuation of nuclear factor-kappaB (NF-kappaB) in inflammation signaling [63].

While the data regarding the effectiveness of curcumin supplementation in protection against the development and progression of NAFLD in preclinical models are encouraging, the translation of these results has not been consistent (Table 1). Recent meta-analyses of 8–9 randomized controlled trials found a positive effect of curcumin supplementation on liver enzymes, serum cholesterol, serum insulin and HOMA-IR, and waist circumference in patients with NAFLD [64,65]. However, several clinical trials directly examining the effectiveness of curcumin supplementation in NAFLD patients have not found any significant benefit for curcumin supplementation. Saadati et al. examined the effects of curcumin supplementation on inflammatory and hepatic fibrosis markers in 50 NAFLD patients and failed to document any improvement in hepatic steatosis or fibrosis with curcumin supplementation [66]. Likewise, a similar study examining the effect of curcumin supplementation on DNA damage caused by increased oxidative stress in NAFLD patients failed to demonstrate any significant protective effect of curcumin supplementation [67]. Lastly, a mixture of natural ingredients, including curcumin, could not document any significant beneficial effect on liver enzymes or metabolic and inflammatory variables in NAFLD patients [68]. The beneficial effects of curcumin supplementation in animal models of NAFLD have yet to be fully observed in clinical trials of NAFLD patients. Clinical trials better designed with different formulations of curcumin [60] may be needed in order to determine if curcumin supplementation is beneficial for NAFLD patients. 

### 3.2. Pomegranate Seed Oil

Pomegranates and pomegranate seed oil may be beneficial for combating obesity since they contain numerous antioxidants such as ascorbic acid, polyphenols, punicic acid and tannins [85]. Pomegranate seed oil (PSO) contains a dense fraction of these antioxidants and is preferable to pomegranate juice, which contains high levels of sugars such as fructose (6.83/100 g) and glucose (6.66/100 g) [86]. The sugars in pomegranate juice such as fructose have been demonstrated to induce obesity, resulting in inflammatory and oxidative stress, both of which promote NAFLD and metabolic syndrome [87]. The high level of sugars found in pomegranate juice may be responsible for the lack of effect on insulin secretion and sensitivity in a clinical trial of patients with obesity [88]. PSO has a greater therapeutic potential because it lacks the sugars found in pomegranate juice and contains more natural antioxidant compounds. 

Studies in mouse models of high-fat diet-induced obesity have demonstrated a protective effect of pomegranate seed oil (PSO) on weight gain and body composition [69,70]. PSO supplementation also improved insulin sensitivity in the periphery but did not have any effect on improving liver insulin sensitivity [69,70]. In a double-blind, placebo-controlled, randomized, 16 week clinical study of non-diabetic, obese premenopausal women with and without NAFLD, PSO mixed with brown marine algae fucoxanthin resulted in a significant reduction of body weight, liver fat content, liver enzymes, serum triglycerides, and C-reactive protein in the NAFLD group [71]. A recent study in a mouse model of high-fat diet-induced obesity and NAFLD demonstrated that PSO supplementation reduced hepatic steatosis and fibrosis, improved hepatic mitochondrial function as well as decreasing fasting glucose levels [72]. The improved hepatic steatosis and insulin sensitivity was due to an upregulation of hepatic HO-1 by PSO supplementation [72]. Hepatic HO-1 induction also resulted in an increase in mitofusin-2 (Mfn2), mitochondrial dynamin like GTPase (OPA-1), PR domain containing 16 (PRDM 16), and peroxisome proliferator-activated receptor gamma coactivator 1-alpha (PGC1α) levels as well as the upregulation of hepatic insulin receptor phosphorylation [72]. These effects of HO-1 induction on PGC1α levels in the liver are similar to those observed in the heart [89]. However, the importance of HO-1 induction to the beneficial effects of PSO needs to be evaluated in future studies (Table 1). 

### 3.3. Milk Thistle Seed Oil

The milk thistle plant (*Silybum marianum)* can be processed using several methods, that include ethanol extraction and cold pressing to make milk thistle oil (MTO). MTO contains various polyphenols such as, *p*-coumaric, vanillic acid, and silybin as well a high levels of α-tocopherol [90,91]. Ethanol extraction of milk thistle increases the level of silymarin. Silymarin treatment protects against chemical-induced liver injury as well as numerous liver diseases associated with increased levels of oxidative stress and hepatic steatosis [92,93]. Recently, Pais et al. demonstrated the protective effects of a patented extract of milk thistle (ETHIS-094) in a mouse model of NAFLD to NASH progression [73]. 

Isolated silymarin flavonoids, 2,3-dehydrosilybins A and B decrease lipoperoxidation in the liver through downregulation of hepatic UGT1A1 and subsequent increases in hepatic and plasma bilirubin levels [94]. Thus, it appears that increases in serum bilirubin levels could mediate some of the hepato-protective effects of silymarin. Bilirubin protects against the development of NAFLD in both animal models and human patient populations [95,96,97,98,99]. Bilirubin can protect the liver by acting as a hormone capable of activating nuclear receptors such as PPARα, which can increase the expression of genes that regulate hepatic β-oxidation [32,33,100].

MTO contains very little silymarin when extracted as a cold press oil; however, it retains its polyphenols, free fatty acids, and other natural antioxidants which can be beneficial in the context of NAFLD. MTO is a potent inducer of HO-1 in both adipose tissue and liver [74]. MTO supplementation in dietary-induced obese mice prevented several of the cardiovascular and metabolic pathologies in dietary obesity such as excessive weight gain, hyperglycemia, and hypertension and also decreased obesity-induced hepatic fibrosis through decreases in matrix metalloproteinase-2 (MMP2) and 9 [74]. Inflammation is very prevalent in obesity and drives cardiovascular and metabolic diseases associated with obesity [101]. MTO supplementation decreased obesity-induced hepatic inflammation through the down-regulation of several critical inflammatory mediators such as nephroblastoma overexpressed (NOV), TWIST, and nuclear factor kappa-light-chain-enhancer of activated B cells (NF-kB) [74]. MTO also improved hepatic insulin signaling through increases in PGC-1α and the activation of the β subunit of the insulin receptor (IR) [74]. MTO supplementation attenuates many of the complications associated with obesity and could prove to be a novel dietary treatment for obesity and its cardiovascular and metabolic complications (Table 1). 

### 3.4. Cold Pressed Nigella Sativa Oil

*Nigella Sativa* (*N.Sativa)* is an annual flowering plant with a long history of medicinal use, especially in the Middle Eastern Mediterranean region. The seeds of *N. sativa* contain thymoquinone (TQ) which are present at high levels in essential oils derived from the seeds. *N. sativa* oil and are protective in several different models of liver injury, including aluminum chloride-induced injury, carbon tetrachloride (CCL4) induced liver injury, and carboplatin-induced liver damage [102,103,104]. *N. sativa* oil supplementation decreases serum markers of liver injury such as aspartate transaminase, alanine transaminase, alkaline phosphatase, and lactate dehydrogenase. Increased levels of oxidative stress are typical hallmarks of all of these different hepatotoxic chemicals, and *N. sativa* oil supplementation was able to decrease reactive oxygen species levels as well as boost the levels of antioxidants such as glutathione peroxidase and superoxide dismutase [102,103,104].

*N. sativa* oil and TQ are potent inducers of HO-1 in the lung, heart, liver, and cell lines such as HaCaT cells derived from human keratinocytes [75,76,105,106,107]. Recent studies have demonstrated that supplementation of TQ alone or given in combination with fish oil omega-3 prevents the development of dietary obesity-induced NAFLD [75,76]. TQ alone or in combination, also alleviated other dietary-induced obesity complications such as hypertension, decreases in oxygen consumption, and insulin resistance [75,76]. The ability of TQ supplementation to lower blood pressure in dietary-induced obesity is similar to a previous study demonstrating the antihypertensive actions of TQ supplementation in a model of nitric oxide deficient hypertension [106]. TQ supplementation also restored the levels of important markers of mitochondrial functions such as mitofusin-1 (MFN-1), MFN-2, and OPA1 in the liver [75,76]. Dietary obesity also induced hepatic fibrosis and inflammation, both of which were significantly reversed by TQ supplementation [75,76]. TQ supplementation also increased hepatic Akt signaling leading to alterations in insulin receptor phosphorylation and liver insulin resistance improvements [76]. Rashidmayvan et al. recently tested the effectiveness of *N. sativa* oil in NAFLD patients in an eight week randomized, double-blind, placebo-controlled clinical trial [77]. Reductions in fasting blood glucose, lipid profiles (TG, TC, LDL, VLDL), liver enzymes (aspartate transaminase (AST) and alanine transaminase (ALT)), and inflammatory markers such as high-sensitivity C-reactive protein (hs-CRP), interleukin-6 (IL-6), tumor necrosis factor-α (TNFα), were found as a result of consumption of NS seed oil [77]. NS see oil also increased HDL levels, compared to the placebo group [77]. Thus, it appears that *N. sativa* oil either by itself or used besides other dietary supplements such as omega-3 fish oil may help protect against NAFLD in human patient populations. Further studies are needed to determine the specific role of HO-1 induction in the protection afforded by *N. sativa* oil supplementation (Table 1). 

### 3.5. Resveratrol

Resveratrol is a phytoalexin, a class of compounds produced by many plants and is found in high levels in red wine due to the fact that this type of wine is often fermented with the skins of the grapes, which contain high levels of resveratrol. Resveratrol is also found in berries, including blueberries, raspberries, mulberries, cranberries, and bilberries. Resveratrol induces HO-1 in the brain, heart, kidney, and liver. In the brain, resveratrol induces HO-1 to protect against ischemic stroke [108,109]. Resveratrol also protects the heart in models of diabetic-induced cardiomyopathy as well as ischemia-reperfusion injury [110,111]. Resveratrol induces HO-1 through activation of the Nrf2 pathway in both cultured PC12 cells and in Hep G2 liver cells through activation of the extracellular signal-regulated kinase (ERK) pathway [112,113]. Resveratrol also acts epigenetically to alter methylation of the Nrf2 promoter in the liver as treatment of HepG2 cells with 5-aza, a demethylating agent, prevents resveratrol induced Nfr2 gene induction and subsequent increases in HO-1 [114]. 

Resveratrol exerts protective effects in numerous preclinical models of NAFLD through a variety of mechanisms. Zucker fatty rats treated with resveratrol for six weeks demonstrated decreased liver weight and triglyceride content [78]. Resveratrol treatment also reduced hepatic oxidative stress and increased hepatic carnitine palmitoyltransferase-Ia (CPT-Ia) and acyl-coenzyme A oxidase (ACO) activity [78]. In a rat model of dietary-induced NAFLD, resveratrol treatment decreased NAFLD severity through decreases in inflammatory cytokines such as TNFα, and through increases in superoxide dismutase, glutathione peroxidase, and catalase [79]. In Hep G2 hepatocytes, resveratrol treatment has been demonstrated to have a profound effect on gene expression upregulating the expression of Sirt1 and forkhead box O1 (FOXO1), and downregulating the expression of sterol regulatory element-binding protein1 (SREBP1) [80]. In separate studies, Izdebska et al. demonstrated that resveratrol treatment alleviated high glucose-induced steatosis through increases in mitochondrial activity [81]. Selective targeting of hepatocytes for gene delivery can be achieved with molecules that recognize and bind to receptors specifically found on the surface of hepatocytes. One class of molecules that can be used for hepatocyte-specific targeting are asialoglycoprotein receptors (ASGPR). ASGPRs are receptors overexpressed in hepatic parenchymal cells and not expressed in non-hepatic cells. A recent study delivered resveratrol in vivo by utilization of lysozyme micelles (Lys M) coated with D- (+) galactose (Gal) conjugated to an oxidized starch (Gal-OS) polymer [82]. The targeted resveratrol attenuated hepatic steatosis and improved hepatic insulin sensitivity in dietary-induced obese NAFLD mice through modulation of AMPK/SIRT1/FAS/SREBP1c signaling pathway [82]. Resveratrol has also been used in combination with another natural product, HO-1 inducer, curcumin, to protect against high fat diet-induced NAFLD [115]. Resveratrol supplementation, in addition to exercise training, improved hepatic steatosis, and decreased hepatic inflammation, and increased antioxidant capacity in the liver [116]. 

Clinical trials of resveratrol supplementation have not been as promising as those of preclinical models (Table 1). In small randomized, placebo-controlled clinical trials of NAFLD and control patients, resveratrol supplementation had minimal effect on hepatic steatosis, insulin sensitivity, plasma markers of inflammation, and antioxidants [83,84]. Kantartzis et al. reported in a large randomized, placebo-controlled clinical trial with over 100 NAFLD patients that resveratrol supplementation failed to have any measurable effect on hepatic steatosis or other cardiometabolic risk factors [117]. A similar study on a smaller group of patients was unable to find any protective effect of resveratrol supplementation on atherogenic indices and blood pressure in patients with NAFLD [118]. Likewise, results from several meta-analysis studies examining the effectiveness of resveratrol supplementation in NAFLD patients failed to support resveratrol in the management of NAFLD and its metabolic and cardiovascular complications [119,120]. Overall, while the data supporting the beneficial effects of resveratrol in NAFLD treatment in preclinical models is strong, clinical trials have not shown similar promise. It is possible that resveratrol could prove to be beneficial to NAFLD patients when used in concert with other natural products or with lifestyle modifications. 

## 4. Conclusions

In NAFLD, levels of HO-1 are decreased, possibly through alterations in the levels of BACH1, leading to increased ROS production and inflammation, both of which contribute to enhanced triglyceride storage and reductions in glycogen (Figure 1). Natural product HO-1 inducers like curcumin, PSO, MTO, *N. sativa* oil, and resveratrol activate HO-1 expression to reduce ROS production and inflammation. Activation of HO-1 also stimulates PPARα, which results in the burning of hepatic fatty acids and increases in glycogen storage. HO-1 induction also plays a key role in the transition of macrophages from the inflammatory M1 phenotype into the anti-inflammatory M2 [121,122]. Whether used alone or in combination with other dietary supplements or medications, natural product HO-1 inducers may represent novel treatment options for patients most at risk for developing NAFLD and its complications. 

## Figures and Tables

**Figure 1 ijms-21-09493-f001:**
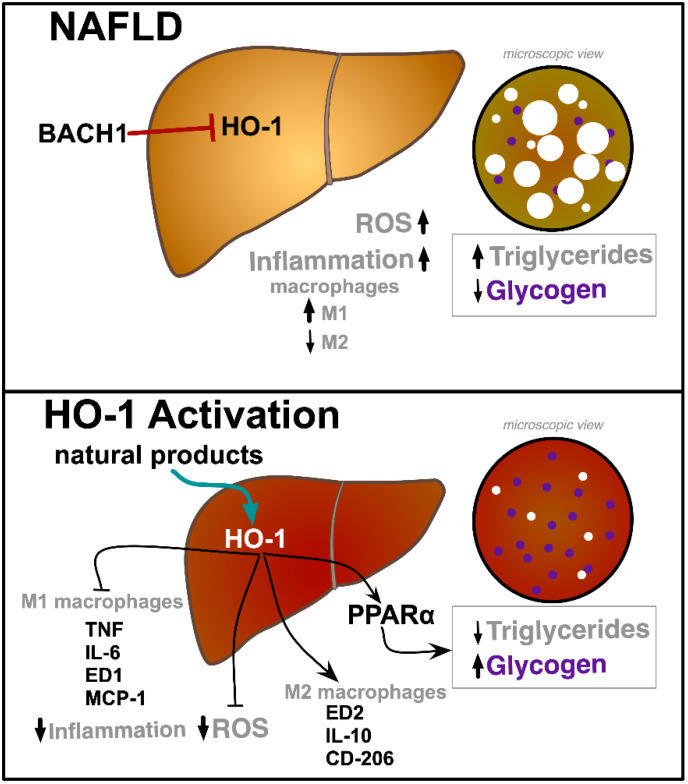
Schematic diagram on the potential role of natural product HO-1 inducers in NAFLD. In NAFLD, HO-1 levels are suppressed by several mechanisms, including increased BACH1 activity. This increases reactive oxygen species (ROS) levels as well as macrophage M1 polarization as opposed to M2, both of which contribute to increasing triglyceride accumulation and reductions in glycogen content. Natural product induction of HO-1 decreases M1 macrophage levels and their resultant inflammatory cytokines, including tumor necrosis factor (TNF), interleukin-6 (IL-6), cluster of differentiation 68 (ED1), monocyte chemoattractant protein-1 (MCP-1), and at the same time increases M2 macrophage levels and their resultant anti-inflammatory cytokines, ED2, interleukin-10 (IL-10), and mannose receptor (CD-206). HO-1 also increases the activity of peroxisome proliferator-activated receptor-α (PPARα) to decrease hepatic triglyceride levels and increase hepatic glycogen content. ↑ indicate “increase” and ↓ indicate “decrease” respectively. The white circles represent fat droplets and the purple circles represent glycogen deposits in the liver.

**Table 1 ijms-21-09493-t001:** Summary of preclinical and clinical studies of natural product HO-1 inducers on nonalcoholic fatty liver disease (NAFLD). HFD, high fat diet; Gal-OSL/Res- D- (+) galactose- oxidized starch-lysozyme/resveratrol.

Product	Model/Administration	Effect	Reference
Curcumin	Wistar Rats/~100 mg/kg body weight/day	Protective against NAFD & NASH	[59]
	C57BL/6J mice HFD/Pegylated 50–100 mg/kg	Protective against NAFD & NASH	[60]
	Meta-analysis in NAFLD Patients/> /< 500 mg/day	Favorable effect on metabolic markers	[64]
	Meta-analysis in NAFLD Patients/70 to 3000 mg/day	Positive effect on visceral fat and abdominal obesity	[65]
	50 NAFLD Patients/1500 mg/day, 12 weeks	No effect on steatosis, slight effect on fibrosis	[67]
	113 NAFLD Patients/35 mg/day, 3 months	No effect on steatosis or metabolic parameters	[68]
Pomegranate Seed Oil	Cd-1 mice HFD/61.79 mg/day, 14 weeks	Lowered body weight and improved insulin sensitivity	[69]
	C57BL/6J mice HFD/1% wt:wt in diet, 12 weeks	Ameliorated high-fat diet induced obesity and insulin resistance	[70]
	NAFLD and control patients/200–300 mg/day, 16 weeks	Lowered hepatic steatosis and improved liver function	[71]
	C57BL/6J mice HFD/40 mL/kg food, 8 weeks	Protective against hepatic steatosis and fibrosis	[72]
Milk Thistle Seed Oil	C57BL/6J mice HFD/500–1000 mg/kg, 4 weeks	Protective against NAFD & NASH	[73]
	C57BL/6J mice HFD/2%/kg/day, 8 weeks	Attenuation of hepatic steatosis, inflammation, and insulin resistance	[74]
Cold Pressed Nigella Sativa Oil	C57BL/6J mice HFD/3%, 8 weeks	Attenuation of hepatic steatosis, induction of HO-1 and improved mitochondrial function	[75]
	C57BL/6J mice HFD/3%, 8 weeks	Attenuation of hepatic steatosis, induction of HO-1 and improved mitochondrial function	[76]
	44 NAFLD Patients/1 g/day oral, 8 weeks	Improved lipid profile, markers of inflammation, and liver enzymes	[77]
Resveratrol	Zucker Fatty Rats/15 mg/kg body weight/day	Reduced hepatic steatosis and oxidative stress	[78]
	Wistar Rats (a high carbohydrate-fat free modified diet)/10 mg/day oral	Reduced hepatic steatosis, oxidative stress, and inflammation	[79]
	HepG2 Cells/0.2–40 μM	Reduced lipid accumulation, increased Sirt1-Fox0, improved mitochondrial function	[80,81]
	C57BL/6J mice HFD/Gal-OSL/Res 200 mg/kg i.v. every other day	Protective against NAFLD	[82]
	28 NAFLD patients/1500 mg/day, 8 weeks	No effect on NAFLD	[83]
	20 Male NAFLD patients	No effect on NAFLD	[84]

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
