# Peer review of "Natural Product Heme Oxygenase Inducers as Treatment for Nonalcoholic Fatty Liver Disease"

_ijms, 2020, doi:10.3390/ijms21249493_

Round 1
Reviewer 1 Report
Comments
The present review points on beneficial effects of dietary supplements such as curcumin, pomegranate seed oil, milk thistle oil, Nigella Satvia oil, resveratrol etc., for the nonalcoholic fatty liver disease (NAFLD) treatment. Authors concluded that beneficial actions of these compounds are associated with the induction of heme oxygenase-1 (HO-1) and its different physiological activities, and the targeting HO-1 through dietary supplements may be a useful therapeutic for NAFLD. The paper is well written and the concept is nice. Still there are several points to address.
It is well-known, that although mostly beneficial, the HO reaction can also produce deleterious effects, predominantly attributed to excessive product formation. Underrated so far is, however, that HO-1 may exert effects additionally via modulation of the cellular heme levels. Heme, besides being an often-quoted generator of oxidative stress, plays also an important role as a signaling molecule. Heme controls the anti-oxidative defense, circadian rhythms, activity of ion channels, glucose utilization, erythropoiesis, and macrophage function. This broad spectrum of effects depends on its interaction with proteins ranging from transcription factors to enzymes. In degrading heme, HO has the potential to exert effects also via modulation of heme-mediated pathways.
Therefore, to my mind, there are not only beneficial effects, as mentioned by the authors in the review. It would be very important and to discuss the dual role of HO-1 and to show that the dietary supplements do not have side effects due to suppression of HO-1.
There are few figures and tables in the present review. It would be nice to combine all information on dietary supplements and their effects on NAFLD in the Table.
Author Response
Comment #1 “It is well-known, that although mostly beneficial, the HO reaction can also produce deleterious effects, predominantly attributed to excessive product formation. Underrated so far is, however, that HO-1 may exert effects additionally via modulation of the cellular heme levels. Heme, besides being an often-quoted generator of oxidative stress, plays also an important role as a signaling molecule. Heme controls the anti-oxidative defense, circadian rhythms, activity of ion channels, glucose utilization, erythropoiesis, and macrophage function. This broad spectrum of effects depends on its interaction with proteins ranging from transcription factors to enzymes. In degrading heme, HO has the potential to exert effects also via modulation of heme-mediated pathways.”
Response: In response to this comment we have included a discussion about the physiological roles of heme please see lines 45-58 and 53-54 on page 2.
Comment #2- “Therefore, to my mind, there are not only beneficial effects, as mentioned by the authors in the review. It would be very important and to discuss the dual role of HO-1 and to show that the dietary supplements do not have side effects due to suppression of HO-1.”
Response: We have highlighted the effects of each of the dietary supplements on HO-1 to highlight that they do not suppress HO-1 activity.
Comment #3- “There are few figures and tables in the present review. It would be nice to combine all information on dietary supplements and their effects on NAFLD in the Table.”
Response: In direct response to this comment by the reviewer, we have included a new Table (Table 1) which highlights the information of the dietary supplements and their effects on NAFLD
Reviewer 2 Report
Comments to the authors
I have read with interest the review by Stec and Hinds entitled “Natural Product Heme Oxygenase Inducers as Treatment for Nonalcoholic Fatty Liver Disease”
In the current review the authors highlight the role of oxidative stress in driving the progression of non-alcoholic fatty liver disease. More specifically, they explain the protective effects of HO-1, one of the main enzymes responsible for antioxidant activity, against liver damage. The focus of this review is on the significance of a targeted induction of HO by dietary supplements as a potential therapeutic strategy against fatty liver disease and the results of a number of clinical and preclinical studies with natural product HO-inducers are summarized.
I agree with the authors in the importance of oxidative stress in driving NAFLD, the need for the development of new therapies, and the potential benefit of natural product HO-1 inducers as supplementary therapy. The only difficulty in interpretation of the summarized studies is that these natural products will not only induce HO-1 but also additional anti-oxidative targets like e.g. a number of Nrf2-regulated target genes, which makes it difficult to ascribe the beneficial outcome exclusively to HO-1 induction. In additions, Nrf2-activation does not always cause protective effects, nicely reviewed under https://doi.org/10.1016/j.phrs.2020.104760. The authors should mention that somewhere in the text. Overall, this is a well written comprehensible review.
Merely some minor corrections because of missing punctuation marks, missing words or incorrect sentences are needed e.g. in line 46 (point), line 96 (by), line 117 (sentence), line 145 (placed), line 156 (which), line 263 (sentence), line 307 (sentence), line 336 (decreased), line 355 (point), line 28 and 30 (Bilirubin). In addition, some abbreviations are not explained and should be added, like e.g. AMPK, MFN, OPA, CYP, PRDM,UGT, MMP, NOV, TWIST, SIRT and some are not explained at the first used like SREBP and PGC. Concerning the reference list, in most references the DOI is given, but not in all, this should be adjusted.
Author Response
Comment #1-“I agree with the authors in the importance of oxidative stress in driving NAFLD, the need for the development of new therapies, and the potential benefit of natural product HO-1 inducers as supplementary therapy. The only difficulty in interpretation of the summarized studies is that these natural products will not only induce HO-1 but also additional anti-oxidative targets like e.g. a number of Nrf2-regulated target genes, which makes it difficult to ascribe the beneficial outcome exclusively to HO-1 induction. In additions, Nrf2-activation does not always cause protective effects, nicely reviewed under https://doi.org/10.1016/j.phrs.2020.104760. The authors should mention that somewhere in the text. Overall, this is a well written comprehensible review.”
Response: We thank the reviewer for the comment regarding the review. In response to this comment, we have added a brief discussion regarding the controversial role of NRF2 in NAFLD please see lines 145-162 pages 4-5 as well as reference 52.
Comment #2- “Merely some minor corrections because of missing punctuation marks, missing words or incorrect sentences are needed e.g. in line 46 (point), line 96 (by), line 117 (sentence), line 145 (placed), line 156 (which), line 263 (sentence), line 307 (sentence), line 336 (decreased), line 355 (point), line 28 and 30 (Bilirubin). In addition, some abbreviations are not explained and should be added, like e.g. AMPK, MFN, OPA, CYP, PRDM,UGT, MMP, NOV, TWIST, SIRT and some are not explained at the first used like SREBP and PGC. Concerning the reference list, in most references the DOI is given, but not in all, this should be adjusted.”
Response: We thank the reviewer for pointing out these grammatical errors in our manuscript and have corrected each of them according to the reviewer’s comments.
Round 2
Reviewer 1 Report
Dear Authors,
Thank you very much for careful correction of you reviewer paper and addressing all my comments. I think the paper now could be accepted for publication.
Kind regards,